# Covid-19 vaccine effectiveness against general SARS-CoV-2 infection from the omicron variant: A retrospective cohort study

Lior Rennert[1]*, Zichen Ma[2], Christopher S. McMahan[3], Delphine Dean[4]

1 Department of Public Health Sciences, Clemson University, Clemson, South Carolina, United States of America, 2 Department of Mathematics, Colgate University, Hamilton, New York, United States of America, 3 School of Mathematical and Statistical Sciences, Clemson University, Clemson, South Carolina, United States of America, 4 Department of Bioengineering, Clemson University, Clemson, South Carolina, United States of America

* liorr@clemson.edu

**Data Availability Statement:** The raw data are protected and are not available due to data privacy laws. Covid-19 data used in this study can be

## Abstract

We aim to estimate the effectiveness of 2-dose and 3-dose mRNA vaccination (BNT162b2 and mRNA-1273) against general Severe Acute Respiratory Syndrome Coronavirus 2 (SARS-CoV-2) infection (asymptomatic or symptomatic) caused by the omicron BA.1 variant. This propensity-score matched retrospective cohort study takes place in a large public university undergoing weekly Coronavirus Disease 2019 (Covid-19) testing in South Carolina, USA. The population consists of 24,145 university students and employees undergoing weekly Covid-19 testing between January 3rd and January 31st, 2022. The analytic sample was constructed via propensity score matching on vaccination status: unvaccinated, completion of 2-dose mRNA series (BNT162b2 or mRNA-1273) within the previous 5 months, and receipt of mRNA booster dose (BNT162b2 or mRNA-1273) within the previous 5 months. The resulting analytic sample consists of 1,944 university students (mean [SD] age, 19.64 [1.42] years, 66.4% female, 81.3% non-Hispanic White) and 658 university employees (mean [SD] age, 43.05 [12.22] years, 64.7% female, 83.3% non-Hispanic White). Booster protection against any SARS-CoV-2 infection was 66.4% among employees (95% CI: 46.1–79.0%; *P* < .001) and 45.4% among students (95% CI: 30.0–57.4%; *P* < .001). Compared to the 2-dose mRNA series, estimated increase in protection from the booster dose was 40.8% among employees (*P* = .024) and 37.7% among students (*P* = .001). We did not have enough evidence to conclude a statistically significant protective effect of the 2-dose mRNA vaccination series, nor did we have enough evidence to conclude that protection waned in the 5-month period after receipt of the 2nd or 3rd mRNA dose. Furthermore, we did not find evidence that protection varied by manufacturer. We conclude that in adults 18–65 years of age, Covid-19 mRNA booster doses offer moderate protection against general SARS-CoV-2 infection caused by the omicron variant and provide a substantial increase in protection relative to the 2-dose mRNA vaccination series.

requested at the following link: https://www.clemson.edu/covid-19/testing/research-data.html.

**Funding:** LR and DD acknowledge support from US National Institutes of Health (P20 GM121342) during conduct of this study. DD acknowledges support from the State of South Carolina (CARES act) for building of the on-campus Covid-19 testing lab and support from US National institutes of Health (R01 MH111366) during conduct of this study. LR, ZM, and CM acknowledge salary support from Clemson University for consulting and modelling work pertaining to development and evaluation of public health strategies (project #1502934). The funders had no role in study design, data collection and analysis, decision to publish, or preparation of the manuscript.

**Competing interests:** The authors have declared that no competing interests exist.

## Introduction

Severe Acute Respiratory Syndrome Coronavirus 2 (SARS-CoV-2), the virus that causes Coronavirus Disease 2019 (Covid-19), was first reported in December of 2019 in Wuhan City, China [1]. As of December 6th, 2022, the World Health Organization has reported over 6.4 million cases of Coronavirus Disease 2019 (Covid-19) [2]. SARS-CoV-2 is primarily transferred through droplets and aerosol particles continating the virus [3]. The B.1.1.529 (omicron) variant of SARS-CoV-2, first detected in South Africa in November 2021, is more infectious than any previous variant of SARS-CoV-2 [4]. Omicron has caused a record number of daily SARS-CoV-2 infections in the United States (US) and world wide [5]. While the Covid-19 mRNA vaccines (BNT162b2 or mRNA-1273) have shown initial strong protection against previous SARS-CoV-2 variants [6–8], protection against symptomatic infection quickly declines [6, 9]. Furthermore, several studies have shown that both the 2-dose and 3-dose mRNA vaccination series offer less protection against the omicron variant [10–12]. Additional studies have demonstrated Covid-19 vaccines have lower neutralization efficacy against omicron and are less protective against hospitalization [10, 13]. However, data on the extent to which Covid-19 vaccines are protective against general infection from omicron, including asymptomatic, presymptomatic, and symptomatic infection, is limited.

While SARS-CoV-2 infections caused by the omicron variant are associated with a reduced risk of severe disease compared to previous variants [14], high infection rates increase the risk of severe disease for the population as a whole. Protection against any SARS-CoV-2 infection is therefore important for reduction of both individual and community risk. In this study, we assess effectiveness of the 2-dose and 3-dose messenger RNA (mRNA) vaccination series approved by the U.S. Food and Drug Administration (FDA), BNT162b2 (Pfizer-BioNTech) and mRNA-1273 (Moderna), in a large public university student and employee population undergoing mandatory weekly testing during January 2022. Because such screening captures all infections (including asymptomatic and pre-symptomatic), this study setting is ideal for evaluation of protection against general SARS-CoV-2 infection.

## Methods

### Study design and population

The study population consists of students and employees undergoing mandatory testing between January 3rd and January 31st of 2022 at Clemson University in South Carolina (SC). The study sample was restricted to young-adult students between 18 and 24 years and university employees between 18 and 64 years. Exclusion criteria is provided in Fig 1, and consists of student athletes, recipient of a vaccine dose other than BNT162b2 or mRNA-1273, individuals receiving the second dose of BNT162b2 less than 21 days after their first dose or second dose of mRNA-1273 less than 28 days after first dose, recipient of a booster dose less than 5 months after the second dose of the mRNA-1273 or BNT162b2, and individuals with invalid vaccination cards. Per surveillance testing protocols, individuals testing positive were not required to partake in mandatory pre-arrival or surveillance testing for 90 days since the original positive test result, and are therefore excluded from analyses [15]. Individuals who have self-reported conditions impacting immune response, including HIV, cancer, lupus, rheumatoid arthritis, or solid organ or bone marrow transplant, and who have self-reported medication use of steroids, chemotherapy, or immunosuppressants, were excluded from the sample. Students vaccinated prior to March 31st, 2021, and employees vaccinated prior to March 8th, 2021, were excluded, since only high-risk individuals were eligible to be vaccinated in SC during this time period.

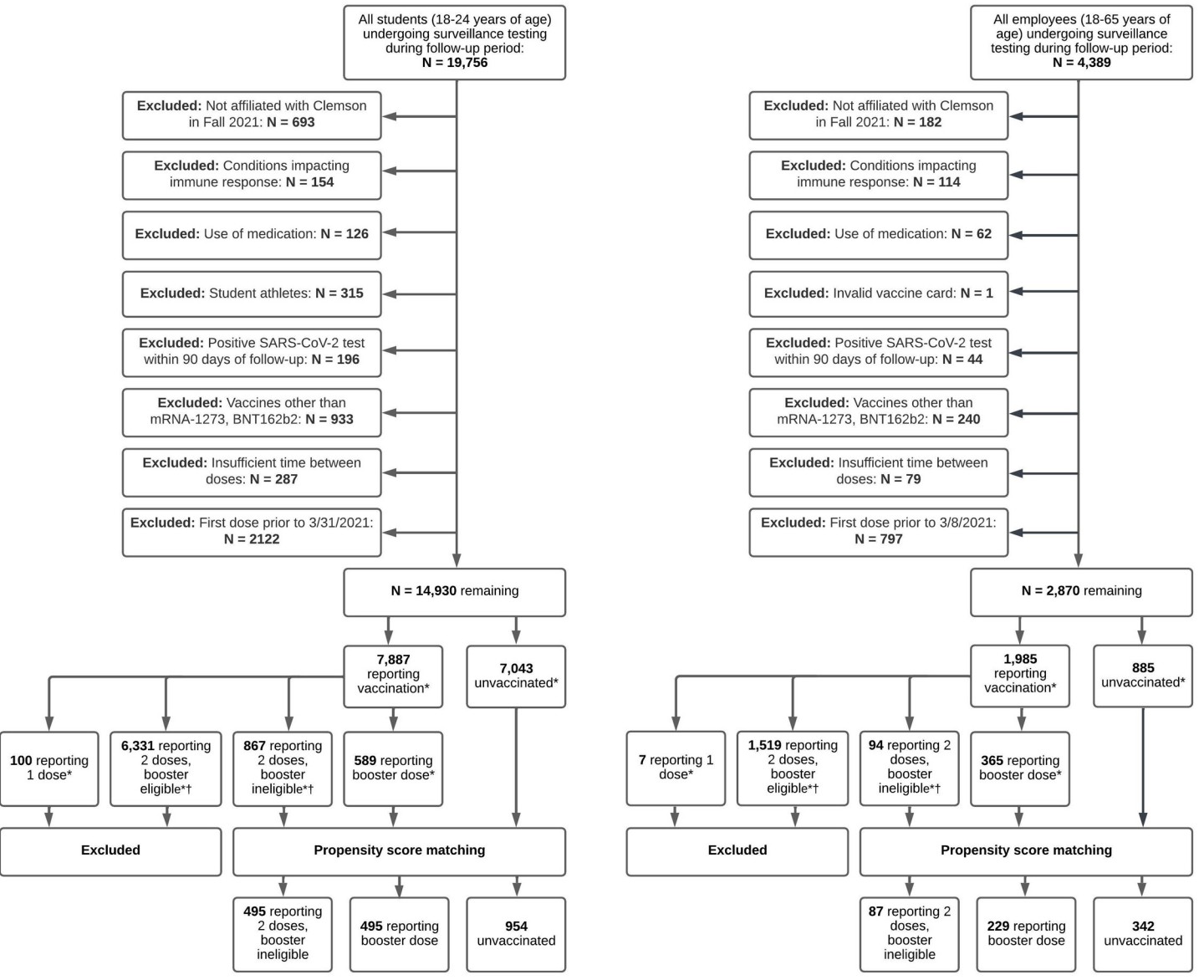

**Fig 1. Flowchart of sample selection process for students and employees in main analysis during follow-up period between January 3rd, 2022 and January 31st, 2022.**
*By end of follow-up (Jan 31st, 2022). † Booster eligible individuals are those receiving their 2nd mRNA dose prior to August 24, 2021. The fully vaccinated, *booster eligible* cohort could consist of a mix of fully vaccinated individuals (completion of 2-dose mRNA series) and boosted individuals (completion of 3-dose mRNA series), and are therefore excluded. The fully vaccinated, *booster ineligible* cohort consists of individuals vaccinated within 5 months of end of follow-up (Jan 31st, 2022) and were therefore not eligible to receive the booster dose.

## Ethics statement

Ethical review for this study was obtained by the Institutional Review Board (IRB) of Clemson University (IRB # 2021-043-02). No consent was needed for this study; students consented to being tested and voluntarily uploaded vaccination information, and we used de-identified data for these analyses.

## Vaccination status

Vaccine information was collected through voluntary upload of vaccination cards through Clemson University's Covid-19 Voluntary Vaccine Upload Tool. Financial incentives were

provided to individuals uploading proof of complete vaccination during the Fall 2021 semester [16]. Individuals also had the option to upload information on booster doses at any point. While this was encouraged through university communications, financial incentives were not offered for booster upload. Vaccination data includes vaccine manufacturer and administration dates of first, second, and booster doses. At Clemson University, the Fall semester typically starts in middle to late-August and continues through mid-December. The spring semester typically starts in early to mid-January and continues through early May.

Individuals who were not affiliated with the University during the Fall 2021 semester were excluded from analyses, since this population was not eligible to participate in the University's financial incentive program for voluntary upload of Covid-19 vaccination and is therefore subject to underreporting. Furthermore, the incentive program only required proof of full vaccination. That is, university students and employees were provided financial incentives to upload proof of completion of 2-dose Covid-19 mRNA vaccine series (BNT162b2 or mRNA-1273) or 1-dose of the Jansen/Johnson and Johnson (Ad26.COV2.S) Covid-19 vaccine during the Fall 2021 semester. Many individuals were not eligible for a booster dose when they provided proof of complete vaccination. Even among those that were booster eligible (defined as completed 2-dose series mRNA series > 5 months ago), there were no incentives to upload proof of the booster dose and thus many students and employees who received a booster dose likely never reported it.

Therefore, in this population, individuals labeled as "booster eligible" consist of a mixed population of those who truly received only 2 doses, but also those who received the booster dose and never reported it. Since is no way of differentiating between these two subgroups, we cannot include individuals that were booster eligible but never reported a booster dose. Otherwise, there would be many boosted individuals classified as only completing full vaccination. Therefore, from the fully vaccinated cohort (i.e., people reporting only a 2-dose mRNA vaccination series), we excluded all individuals who received their 2nd dose more than 5 months prior to the end of the follow-up (Jan 31st, 2022). This ensures that the resulting fully vaccinated cohort would be devoid of boosted individuals since its constituents would not be eligible to receive the mRNA booster dose during the study period.

Vaccination status was coded as partially vaccinated if 14-days had passed since the first dose of mRNA-1273 or BNT162b2, fully vaccinated if at least 14 days had passed since the second dose of BNT162b2 or mRNA-1273 [17], and boosted if 7 days had passed since the receipt of a the BNT162b2 or mRNA-1273 booster dose [18]. The analytic sample was restricted to individuals reporting no vaccination, completion of the 2-dose mRNA vaccination series within 5 months of end of follow-up (fully vaccinated), and receipt of an mRNA booster dose (boosted).

## SARS-CoV-2 testing

Pre-arrival testing was required for all individuals between January 3rd and January 12th (in-person instruction), followed by weekly surveillance testing. Residential students were also required to test upon arrival to residence halls (beginning January 9th). All students had the option of receiving a SARS-CoV-2 test on campus or obtaining their test elsewhere and uploading their result to the University's Covid-19 test upload portal. Accepted methods included nasal, throat, or saliva-based polymerase-chain-reaction (PCR) tests [19]. Testing was available on-campus through the University's *high-complexity CLIA-certified clinical diagnostics lab, which utilized* saliva-based PCR tests. This compromised the majority of tests conducted during the study follow-up period (N = 101,982; 92.9% of all tests). Test results with quantification cycle (Cq) values under 33 were considered positive for SARS-CoV-2 (test

sensitivity $\geq$ 95%, test specificity $\geq$ 99.5%) [20, 21]. Testing was mandated for all university-affiliated individuals; access to campus facilities was restricted until a negative test result was obtained. Therefore, unreported at-home testing is not expected to impact study findings. Pre-arrival and surveillance testing protocols for previous semesters, including clinical descriptions of testing procedures, are described elsewhere [15, 20]. It is estimated that 99.2% of all SARS-CoV-2 infections in South Carolina were caused by the original omicron variant (BA.1) in January of 2022 [22].

### Data collection and management

SARS-CoV-2 testing data was collected by three independent sources: Rymedi (Greenville, SC, USA) for tests conducted by the University CLIA-certified diagnostics lab, which included demographic and clinical information along with university affiliation (collected at test registration) [20], Clemson University Student Health Center for all other tests conducted on campus, and Clemson Computing and Information Technology (CCIT) for tests conducted off-campus and uploaded via the testing upload portal. Vaccine information was collected through CCIT via the voluntary vaccination upload tool. All data was stored and managed on a secure server provide by Clemson University.

### Propensity score matching

Propensity-score matching was used to adjust for confounding of vaccine effectiveness estimates due to reporting bias and differences in health-seeking behavior. To achieve covariate balance while preserving sample size, matching was conducted using a common-referent approach [23]. In the student sample, 1:1 nearest neighbor caliper matching was conducted between boosted and fully vaccinated individuals and 1:2 nearest neighbor caliper matching was conducted between boosted and unvaccinated individuals [24, 25]. The post-matched student sample consists of unvaccinated and fully vaccinated individuals who had a common match with a boosted individual [23]. In the employee study sample, 1:4 nearest neighbor caliper matching was conducted between fully vaccinated and boosted individuals and 1:4 nearest neighbor caliper matching was conducted between fully vaccinated and unvaccinated individuals. The post-matched employee sample consists of boosted and unvaccinated individuals who had a common match with a fully vaccinated individual. The caliper width was set to 0.20 times the standard deviation of the logit of the propensity scores [26]. Variables in the propensity score model included age, sex, affiliation (students: residential or non-residential living, employees: faculty or staff), graduate student status (students only), race/ethnicity, self-reported pre-existing conditions (high blood pressure, heart disease, diabetes, overweight or obesity, kidney disease or dialysis, previous stroke or other neurological condition affecting ability to cough, liver disease, or lung disease), use of tobacco or nicotine products, number of SARS-CoV-2 tests in Fall 2020, Spring 2021, and Fall 2021 semesters, and history of previous SARS-CoV-2 infection.

### Statistical analyses

Demographic and clinical characteristics are provided for the analytic sample and stratified by vaccination status in Tables 1 and 2. Differences between unvaccinated, fully vaccinated, and boosted individuals were assessed using ANOVA for continuous variables and chi-squared tests for categorical variables. Time-varying Cox proportional hazard models were used to estimate the unadjusted hazard rate (HR) and adjusted hazard rate (aHR) of SARS-CoV-2 infection by vaccination status during the follow-up period (1/3/2022-1/31/2022). The outcome was days between start of follow-up and date of first SARS-CoV-2 positive test (event date).

**Table 1. Descriptive characteristics for students in the analytic sample after propensity score matching.**

| Characteristic | Total N = 1944 | Unvaccinated N = 954 | Fully vaccinated[•] N = 495 | Booster N = 495 | P-value |
|---|---|---|---|---|---|
| **Age: Mean (SD)** | 19.64 (1.42) | 19.56 (1.42) | 19.73 (1.43) | 19.70 (1.40) | 0.07 |
| **Race/Ethnicity: N (%)** | | | | | |
| . . .White non-Hispanic | 1580 (81.3%) | 778 (81.6%) | 407 (82.2%) | 395 (79.8%) | 0.59 |
| . . .Black non-Hispanic | 91 (4.7%) | 43 (4.5%) | 24 (4.8%) | 24 (4.8%) | 0.94 |
| . . .Any race Hispanic | 143 (7.4%) | 70 (7.3%) | 34 (6.9%) | 39 (7.9%) | 0.83 |
| . . .All other races non-Hispanic | 130 (6.7%) | 63 (6.6%) | 30 (6.1%) | 37 (7.5%) | 0.67 |
| **Gender: N (%)** | | | | | |
| . . .Female | 1291 (66.4%) | 639 (67.0%) | 330 (66.7%) | 322 (65.1%) | 0.75 |
| . . .Male | 646 (33.2%) | 311 (32.6%) | 164 (33.1%) | 171 (34.5%) | 0.76 |
| . . .Not reported | 7 (0.4%) | 4 (0.4%) | 1 (0.2%) | 2 (0.4%) | 0.79 |
| **Affiliation: N (%)** | | | | | |
| . . .Residential | 985 (50.7%) | 518 (54.3%) | 226 (45.7%) | 241 (48.7%) | 0.005 |
| . . .Non-residential | 959 (49.3%) | 436 (45.7%) | 269 (54.3%) | 254 (51.3%) | 0.005 |
| **Graduate student: N (%)** | 63 (3.2%) | 31 (3.2%) | 17 (3.4%) | 15 (3.0%) | 0.94 |
| **Pre-existing condition: N (%)** [■] | 58 (3.0%) | 30 (3.1%) | 13 (2.6%) | 15 (3.0%) | 0.7 |
| . . .High blood pressure | 9 (0.5%) | 2 (0.2%) | 3 (0.6%) | 4 (0.8%) | 0.45 |
| . . .Heart disease | 1 (0.1%) | 1 (0.1%) | 0 (0.0%) | 0 (0.0%) | 1 |
| . . .Diabetes | 14 (0.7%) | 9 (0.9%) | 1 (0.2%) | 4 (0.8%) | 0.2 |
| . . .Overweight | 36 (1.9%) | 18 (1.9%) | 9 (1.8%) | 9 (1.8%) | 1 |
| . . .Kidney disease | 0 (0.0%) | 0 (0.0%) | 0 (0.0%) | 0 (0.0%) | NA |
| . . .Cough inefficacy | 0 (0.0%) | 0 (0.0%) | 0 (0.0%) | 0 (0.0%) | NA |
| . . .Liver disease | 1 (0.1%) | 1 (0.1%) | 0 (0.0%) | 0 (0.0%) | 1 |
| **Use of tobacco or nicotine products: N (%)** | 37 (1.9%) | 15 (1.6%) | 13 (2.6%) | 9 (1.8%) | 0.24 |
| **SARS-CoV-2 Tests Per Person: Mean (SD)** | 30.61 (11.96) | 29.67 (11.66) | 30.58 (12.15) | 32.45 (12.17) | <0.001 |
| . . .Fall 2020 Semester | 3.62 (4.11) | 3.45 (4.05) | 3.65 (4.10) | 3.92 (4.24) | 0.12 |
| . . .Spring 2021 Semester | 7.30 (7.59) | 6.98 (7.42) | 7.38 (7.53) | 7.84 (7.93) | 0.12 |
| . . .Fall 2021 Semester | 16.20 (3.49) | 16.07 (3.46) | 16.21 (3.25) | 16.45 (3.75) | 0.15 |
| . . .Spring 2022 Semester[¶*] | 3.49 (1.44) | 3.17 (1.35) | 3.34 (1.44) | 4.24 (1.34) | <0.001 |
| **Previous SARS-CoV-2 Infection: N (%)[†]** | 444 (22.8%) | 218 (22.9%) | 114 (23.0%) | 112 (22.6%) | 0.99 |
| **SARS-CoV-2 Infection During Follow-up: N (%)[#]** | 550 (28.3%) | 299 (31.3%) | 171 (34.5%) | 80 (16.2%) | <0.001 |

[•] 2nd dose administered within 5.25 months of study end date

[■] Self-reported presence of any of the following conditions: high blood pressure, heart disease, diabetes, overweight or obesity, kidney disease or dialysis, previous stroke or other neurological condition affecting ability to cough, liver disease, or lung disease; sample size may not add to N due to non-selection of specific conditions.

[¶] During follow-up period (1/3/2022 to 1/31/2022)

[†] Infection occurring prior to follow-up period (1/3/22)

[#] % is proportion of individuals within each population infected with SARS-CoV-2 during follow-up period

[*] Variable not included in propensity score model

Individuals who did not test positive during the follow-up period were right censored at their last negative test date (censoring date). Vaccination status was modeled as a time-varying exposure variable. Because university students tend to engage in high-density social interactions which result in exposure to higher viral loads [27], models were run independently for student and employee populations. We adjust for all potential confounders included in the propensity score models to account for residual imbalance after matching [28]. Specifically, models were adjusted for previous SARS-CoV-2 infection history, age, race, gender, residential status, self-reported pre-existing conditions, use of nicotine or other smoking products, and

**Table 2. Descriptive characteristics for employees in the analytic sample after propensity score matching.**

| Characteristic | Total N = 658 | Unvaccinated N = 342 | Fully vaccinated[•] N = 87 | Booster N = 229 | P-value |
|---|---|---|---|---|---|
| **Age: Mean (SD)** | 43.05 (12.22) | 41.70 (12.61) | 42.97 (11.41) | 45.10 (11.69) | 0.005 |
| **Race/Ethnicity: N (%)** | | | | | |
| . . .White, non-Hispanic | 548 (83.3%) | 289 (84.5%) | 69 (79.3%) | 190 (83.0%) | 0.5 |
| . . .Black, non-Hispanic | 45 (6.8%) | 21 (6.1%) | 8 (9.2%) | 16 (7.0%) | 0.6 |
| . . .All other, races non-Hispanic | 65 (9.9%) | 32 (9.4%) | 10 (11.5%) | 23 (10.0%) | 0.83 |
| **Gender: N (%)** | | | | | |
| . . .Female | 426 (64.7%) | 225 (65.8%) | 58 (66.7%) | 143 (62.4%) | 0.66 |
| . . .Male | 232 (35.3%) | 117 (34.2%) | 29 (33.3%) | 86 (37.6%) | 0.66 |
| **Affiliation: N (%)** | | | | | |
| . . .Faculty | 89 (13.5%) | 39 (11.4%) | 10 (11.5%) | 40 (17.5%) | 0.1 |
| . . .Staff | 569 (86.5%) | 303 (88.6%) | 77 (88.5%) | 189 (82.5%) | 0.1 |
| **Pre-existing condition: N (%)[■]** | 184 (28.0%) | 86 (25.1%) | 23 (26.4%) | 75 (32.8%) | 0.91 |
| . . .High blood pressure | 88 (13.4%) | 44 (12.9%) | 10 (11.5%) | 34 (14.8%) | 0.87 |
| . . .Heart disease | 5 (0.8%) | 3 (0.9%) | 0 (0.0%) | 2 (0.9%) | 0.88 |
| . . .Diabetes | 23 (3.5%) | 7 (2.0%) | 2 (2.3%) | 14 (6.1%) | 1 |
| . . .Overweight | 128 (19.5%) | 57 (16.7%) | 15 (17.2%) | 56 (24.5%) | 1 |
| . . .Kidney disease | 0 (0.0%) | 0 (0.0%) | 0 (0.0%) | 0 (0.0%) | NA |
| . . .Cough inefficacy | 0 (0.0%) | 0 (0.0%) | 0 (0.0%) | 0 (0.0%) | NA |
| . . .Liver disease | 2 (0.3%) | 1 (0.3%) | 0 (0.0%) | 1 (0.4%) | 1 |
| **Use of tobacco or nicotine products: N (%)** | 16 (2.4%) | 8 (2.3%) | 3 (3.4%) | 5 (2.2%) | 0.84 |
| **SARS-CoV-2 Tests Per Person: Mean (SD)** | 27.55 (13.43) | 26.34 (13.84) | 27.39 (13.30) | 29.42 (12.67) | 0.03 |
| . . .Fall 2020 Semester | 1.92 (2.58) | 1.94 (3.01) | 1.74 (1.89) | 1.97 (2.08) | 0.76 |
| . . .Spring 2021 Semester | 10.77 (8.16) | 10.35 (8.21) | 10.52 (8.51) | 11.51 (7.92) | 0.23 |
| . . .Fall 2021 Semester | 10.79 (5.07) | 10.25 (5.39) | 11.01 (4.55) | 11.50 (4.68) | 0.01 |
| . . .Spring 2022 Semester[¶*] | 4.07 (1.61) | 3.81 (1.63) | 4.13 (1.68) | 4.44 (1.48) | <0.001 |
| **Previous SARS-CoV-2 Infection: N (%)[†]** | 110 (16.7%) | 77 (22.5%) | 12 (13.8%) | 21 (9.2%) | <0.001 |
| **SARS-CoV-2 Infections During Follow-up: N (%)[#*]** | 141 (21.4%) | 96 (28.1%) | 23 (26.4%) | 22 (9.6%) | <0.001 |

[•] 2nd dose administered within 5.25 months of study end date

[■] Self-reported presence of any of the following conditions: high blood pressure, heart disease, diabetes, overweight or obesity, kidney disease or dialysis, previous stroke or other neurological condition affecting ability to cough, liver disease, or lung disease; sample size may not add to N due to non-selection of specific conditions.

[¶] During follow-up period (1/3/2022 to 1/31/2022)

[†] Infection occurring prior to follow-up period (1/3/22)

[#] % is proportion of individuals within each population infected with SARS-CoV-2 during follow-up period

[*] Variable not included in propensity score model

number of previous SARS-CoV-2 tests (surrogate for health-seeking behavior) [29, 30]. Unadjusted and adjusted vaccine effectiveness (i.e., protection) was estimated by 1-HR and 1-aHR, respectively; [15] vaccine effectiveness estimates are supplemented with 95% confidence intervals (CI) and p-values (where appropriate). Overall mRNA vaccine protection was estimated via Model 1.1 (S1 Appendix). Effectiveness by vaccine manufacturer was estimated via Model 1.2 (S1 Appendix).

To evaluate vaccine effectiveness by previous infection history, an interaction term between vaccine status and previous infection was included in separate models (S1 Appendix, Models 1.3–1.4). To account for heavy censoring of event times, we apply Firth's penalized likelihood method [31]. Waning vaccine effectiveness over time was evaluated by including time since vaccination as a predictor (S2 Appendix, Models 2.1–2.2). Time-adjusted RR of SARS-CoV-2

infection between the boosted and fully vaccinated groups were estimated via Model 2.3 (S2 Appendix). Additional model details are provided in the S1 and S2 Appendices. All analyses were conducted using R version 4.1.2.

## Results

Descriptive characteristics for students and employees in the post-matching analytic sample are presented in Tables 1 and 2. This sample consists of 1,944 students who were tested for Covid-19 during the follow-up period, of which 954 did not report any vaccination (49.0%), 495 reported full vaccination (25.5%), and 495 reported receiving a booster dose (25.5%). The average age of this population was 19.64 years (SD = 1.42). The majority of the population was non-Hispanic White (81.3%), female (66.4%), lived in residential housing (50.7%), did not report any pre-existing condition (97.0%), did not use tobacco or nicotine products (98.1%), and did not have a previous SARS-CoV-2 infection (77.2%). The average number of SARS-CoV-2 tests per person since the Fall 2020 semester was 30.61 (SD = 11.96). Statistically significant differences were observed in the proportion of residential students (boosted: 48.7%, fully vaccinated: 45.7%, unvaccinated: 54.3%, $P = .005$) and the number of SARS-CoV-2 tests during the Spring 2022 semester (boosted: 4.24, fully vaccinated: 3.34, unvaccinated: 3.17, $P < .001$). However, the latter variable was not included in the propensity score models since it is strongly correlated with the outcome (due to the 90-day exemption from mandatory testing after a positive result). In total, 28.3% of individuals tested positive during the follow-up period. Positivity rates were significantly lower ($P < .001$) among boosted individuals (16.2%) compared to fully vaccinated (34.5%) and unvaccinated individuals (31.3%).

The post-matching analytic sample for employees consists of 658 individuals who were tested for Covid-19 during the follow-up period, of which 342 did not report any vaccination (52.0%), 87 reported full vaccination (13.2%), and 229 reported receiving a booster dose (34.8%). The average age of this population was 43.05 years (SD = 12.22). The majority of the population was non-Hispanic White (83.3%), female (64.7%), staff (86.5%), did not report any pre-existing condition (72.0%), did not use tobacco or nicotine products (97.6%), and did not have a previous SARS-CoV-2 infection (83.3%). The average number of SARS-CoV-2 tests per person since the Fall 2020 semester was 27.55 (SD = 13.43). Statistically significant differences were observed in the individuals' age (boosted: 45.10, fully vaccinated: 42.97, unvaccinated: 41.70, $P = .005$), the number of individuals with a previous SARS-CoV-2 infection (boosted: 9.2%, fully vaccinated: 13.8%, unvaccinated: 22.5%, $P < .001$), the number of SARS-CoV-2 tests during the Fall 2021 semester (boosted: 11.50, fully vaccinated: 11.01, unvaccinated: 10.25, $P = .01$), and the number of SARS-CoV-2 tests during the Spring 2022 semester (boosted: 4.44, fully vaccinated: 4.13, unvaccinated: 3.81, $P < .001$); the latter variable was not included in the propensity score models. In total, 21.4% of individuals tested positive during the follow-up period. Positivity rates were significantly lower ($P < .001$) among boosted individuals (9.6%) compared to fully vaccinated (26.4%) and unvaccinated individuals (28.1%).

Characteristics of the original study populations are presented in S1 Table (students) and S2 Table (employees) and described in S3 Appendix. The matching protocol substantially improved balanced on most variables, as measured by standardized differences pre- and post-matching (S1 and S2 Figs). Using a threshold of standardized differences < 0.25 [32], balance was achieved on 100% and 95.5% of pairwise comparisons between all variables for students and employees, respectively. Using a threshold of 0.10 [28, 32], balance was achieved on 86.3% and 60.0% of pairwise comparisons. For employees, the lower proportion is due to, at least in part, the small sample size for fully vaccinated individuals [28]. Residual imbalance is adjusted for through inclusion of all confounders in the models for vaccine effectiveness [28].

## Vaccine effectiveness

Estimates of vaccine effectiveness for students and employees are presented in Table 3 and Fig 2. The median time between full vaccination and end of follow-up was 4.49 months for students (range: 0.66–5.25) and 4.43 months for employees (range: 0.79–5.21). Protection (adjusted) from the 2-dose mRNA vaccination series was 7.7% among students (95% CI: -11.5–23.5%; $P$ = .409) and 25.6% among employees (95% CI: -17.6–52.9%; $P$ = 0.205). Differences between manufacturers were not statistically significant for students ($P$ = .328) or employees ($P$ = .626).

The median time between booster dose and end of follow-up was 1.31 months for students (range: 0.39–3.51) and 2.03 months for employees (range: 0.52–4.00). Protection (adjusted) from the mRNA booster dose was 45.4% among students (95% CI = 30.0–57.4%; $P$ < .001) and 66.4% among employees (95% CI = 46.1–79.0%; $P$ < .001). Differences between booster manufacturers were not statistically significant for students ($P$ = .652) or employees ($P$ = .417). Furthermore, statistically significant differences were not observed between individuals who mixed and matched (i.e., 2-dose primary series and booster dose from different manufacturers) and those completing the 3-dose BNT16b2 sequence (students: $P$ = .360, employees $P$ = .985), or those completing the 3-dose mRNA-1273 sequence (students: $P$ = .353, employees: $P$ = .589).

Compared to completion of the 2-dose mRNA series, receipt of the mRNA booster dose increased (adjusted) protection by 37.7% among students (95% CI: 15.3–60.2%; $P$ = .001) and 40.8% among employees (95% CI: 5.4–76.2%; $P$ = .024). In a separate analysis accounting for time since vaccination, we directly compare (adjusted) protection among individuals receiving the mRNA booster dose to those completing the 2-dose mRNA series (S2 Appendix, Model

**Table 3. Estimated protection from vaccination against any SARS-CoV-2 infection between January 3rd-31st, 2022.**

| Vaccination Protection | # of Individuals | # Positive (%) | Unadjusted Protection: % (95% CI) | Adjusted Protection: % (95% CI) |
|---|---|---|---|---|
| **Students** | | | | |
| Unvaccinated | 954 | 299 (31.3%) | *Reference* | *Reference* |
| Fully Vaccinated*● | 495 | 171 (34.5%) | 7.6% (-11.6–23.5) [a†] | 7.7% (-11.5–23.5) [a] |
| ...mRNA-1273 | 176 | 53 (30.1%) | 19.6% (-7.7–39.9) [b†] | 17.3% (-10.8–38.3) [b] |
| ...BNT162b2 | 319 | 118 (37%) | 0.5% (-23.3–19.6) [b†] | 2.1% (-21.2–21.0) [b] |
| Booster* | 495 | 80 (16.2%) | 45.1% (29.7–57.1) [a†] | 45.4% (30.0–57.4) [a] |
| ...mRNA-1273 | 199 | 30 (15.1%) | 49.2% (26.2–65.1) [b†] | 48.5% (25.0–64.7) [b] |
| ...BNT162b2 | 296 | 50 (16.9%) | 41.7% (21.3–56.8) [b†] | 42.8% (22.7–57.6) [b] |
| **Employees** | | | | |
| Unvaccinated | 342 | 96 (28.1%) | *Reference* | *Reference* |
| Fully Vaccinated*● | 87 | 23 (26.4%) | 28.0% (-13.6–54.4) [a†] | 25.6% (-17.6–52.9) [a] |
| ...mRNA-1273 | 30 | 10 (33.3%) | 13.1% (-66.1–54.5) [b†] | 14.4% (-64.2–55.4) [b] |
| ...BNT162b2 | 57 | 13 (22.8%) | 34.3% (-17.0–63.1) [b†] | 30.1% (-24.5–60.8) [b] |
| Booster* | 229 | 22 (9.6%) | 67.8% (48.7–79.8) [a†] | 66.4% (46.1–79.0) [a] |
| ...mRNA-1273 | 140 | 16 (11.4%) | 61.4% (34.4–77.3) [b†] | 60.4% (32.4–76.8) [b] |
| ...BNT162b2 | 89 | 6 (6.7%) | 76.0% (46.4–89.3) [b†] | 74.3% (42.1–88.6) [b] |

*Protection is relative to unvaccinated individuals

● 2nd dose administered within 5.25 months of study end date

[a] Estimated via Model 1.1 in S1 Appendix

[b] Estimated via Model 1.2 in S1 Appendix

[†] Estimated via setting $\eta \equiv 0$ in Model 1.1 and 1.2 in S1 Appendix

2.3). Relative to the 2-dose series, the (time-adjusted) risk of SARS-CoV-2 infection for those receiving the mRNA booster dose was 2.16 times lower for students (95% CI: 1.47–3.18, $P < .001$) and 1.52 times lower for employees (95% CI: 0.66–3.46, $P = .324$); however, we did not have enough evidence to conclude statistical significance for the latter effect.

We did not have evidence to conclude that mRNA vaccine protection varied by previous infection history. Significant interactions were not observed between previous SARS-CoV-2 infection and the 2-dose mRNA vaccination series (students: $P = .130$, employees: $P = .184$) or mRNA booster dose (students: $P = .599$, employees: $P = .085$). Months since completion of the 2-dose mRNA vaccination series was not a statistically significant predictor of SARS-CoV-2 infection from Omicron among students (RR = 0.93, 95% CI: 0.86–1.02; $P = .110$) or employees (RR = 0.99, 95% CI: 0.82–1.19; $P = .549$). A significant decline in mRNA booster protection was not observed among students (RR = 1.27, 95% CI: 0.81–1.98; $P = .304$) or employees (RR = 1.03, 95% CI: 0.65–1.65; $P = .886$).

## Discussion

In this study setting, 21.4% of university employees and 28.3% of university students tested positive for SARS-CoV-2 during a 4-week period in which omicron was the dominant SARS-CoV-2 variant. The mRNA booster dose offered moderately high protection for employees (66.4%) and moderate protection for students (45.4%) against any SARS-CoV-2 infection caused by the omicron variant. Furthermore, estimated protection from the mRNA vaccine booster dose was approximately 40% higher for both students and employees relative to the 2-dose mRNA vaccine series.

We did not have enough evidence to conclude that the 2-dose mRNA series was protective against general SARS-CoV-2 infection from the omicron variant. Compared to our study, recent studies have found higher 2-dose and 3-dose mRNA vaccine effectiveness against symptomatic and severe SARS-CoV-2 infection from omicron [10–12]. This is to be expected, since this study evaluates protection against any SARS-CoV-2 infection from

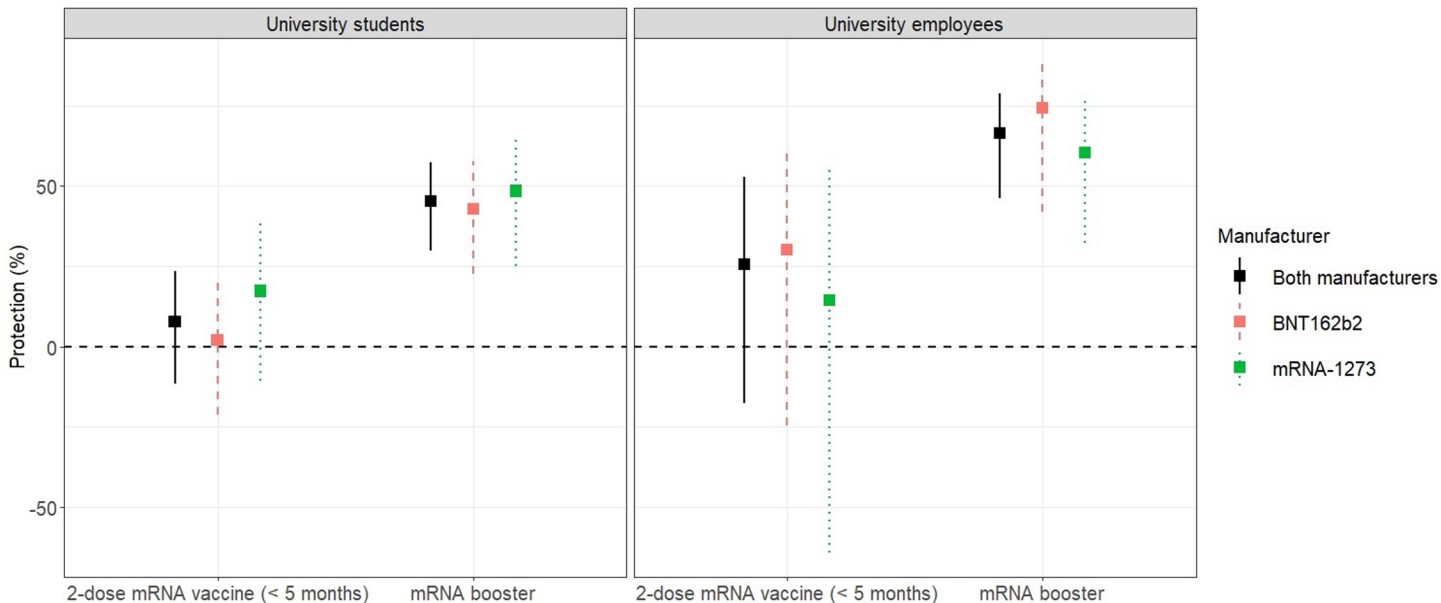

**Fig 2.** Estimates of (adjusted) vaccine protection against any SARS-CoV-2 infection caused by the omicron variant in university students (left) and employees (right).

omicron (asymptomatic, pre-symptomatic, and symptomatic) as opposed to severe infection. While protection against severe SARS-CoV-2 infection is the primary objective of Covid-19 vaccination, protection against general infection is important for mitigating community transmission, since asymptomatic and pre-symptomatic individuals are substantial contributors to disease spread [33]. Therefore, less protection against general infection increases the risk for all individuals in the population. Taken together, this study and others provide evidence that booster shots are beneficial for both individual health and reduction in community transmission [34, 35].

On the surface, our findings of higher protection in older university employees relative to younger university students appear to contradict evidence that Covid-19 vaccines are (relatively) less protective in older adults [36, 37]. However, the risk of infection is dependent on the size of the infecting dose of the virus (i.e., viral load) [38–43]. Compared to the average adult, university students typically engage in more frequent and higher-density social interactions [27, 44]. Such social behavior increases exposure to higher viral loads and subsequently impacts the protective effect of vaccines.

Due to mandated weekly testing, this study design is well-suited for evaluating vaccine effectiveness against any SARS-CoV-2 infection caused by Omicron. Based on the temporal dynamics in SARS-CoV-2 viral shedding [45, 46], most cases were likely detected since the start of January. Mandatory testing also reduces the risk of confounding that may occur in clinical populations due to differences in risk-seeking behavior [47]. Relative to the general population, there is a higher degree of homogeneity in this study sample of university students and employees. This feature, combined with stratification by affiliation (student, employee), propensity score matching, and covariate adjustment based on demographics, occupation, clinical characteristics, and health-seeking behavior, further reduces the risk of confounding in this study.

Our study is subject to several limitations. First, due to low variability in month of vaccination in our study sample and a relatively short follow-up period, there was insufficient power to assess waning vaccine immunity after the $2^{nd}$ and $3^{rd}$ dose against any SARS-CoV-2 infection caused by omicron. Second, the omicron surge in South Carolina began at least one week before arrival testing was initiated [48]. While underreporting of infections prior to this period was less likely for symptomatic individuals (proof of a positive SARS-CoV-2 test provided a 90-day exemption from mandatory testing), asymptomatic individuals during this time-period were less likely to be tested and reported. Because vaccinated individuals are more likely to have mild infections relative to unvaccinated individuals, it is likely that a larger proportion of vaccine breakthrough infections were underreported during this time. While this would lead to overestimation of vaccine effectiveness, the overall impact of underreporting is expected to be low, since only a small fraction of Omicron infections in South Carolina are estimated to have occurred prior to 2022 [48].

Third, misclassification of vaccination status may attenuate estimates of vaccine effectiveness towards the null [17]. Financial incentives offered during the Fall 2021 semester for voluntary vaccination upload likely minimized underreporting of the 2-dose mRNA series. Therefore, unvaccinated individuals are unlikely to be misclassified since the analytic population was restricted to those affiliated with the university during Fall 2021. However, booster doses became widely available after the financial incentive program was announced. It is therefore possible that booster doses are underreported among fully vaccinated individuals (i.e., recipients of the 2-dose mRNA series) who are booster eligible. Underreporting of boosters in this situation would lead to overestimation of 2-dose mRNA vaccine effectiveness among individuals who received their $2^{nd}$ dose more than 5 months prior to the end of study follow-up (i.e., booster-eligible individuals). For this reason, we excluded this population from analysis.

This study evaluated vaccines effectiveness in relatively healthy university populations. Due to our exclusion criteria and matching protocol, the resulting analytic sample for students tended to be both healthier and have greater health-seeking behavior. On the other hand, university students typically have a higher number of social contacts relative to other young adult populations. Hence, this population is more susceptible to contracting SARS-CoV-2 and may be subjected to higher viral loads. Caution must therefore be taken when extrapolating estimates of vaccine effectiveness beyond the analytic population in this study.

## Conclusions

Among adults 18–65 years of age, messenger RNA (mRNA) booster doses offer moderate protection against general SARS-CoV-2 infection caused by the omicron variant. In both university students and employees, receipt of the mRNA booster dose substantially increased protection relative to the 2-dose mRNA vacccination series. The effectiveness of mRNA boosters on both individual health and reduction in community transmission support continued efforts for Covid-19 booster doses. This is especially important in the young-adult population, which has suboptimal vaccine uptake yet is a major contributor to community spread. However, given that approximately one-third to one-half of the study population remains susceptible to breakthrough infection, certain precautions after vaccination may still be warranted.

## Supporting information

**S1 Appendix. Statistical models for vaccine protection ([Table 3]).**
(DOCX)

**S2 Appendix. Statistical models for waning vaccine protection.**
(DOCX)

**S3 Appendix. Study sample description.**
(DOCX)

**S1 Table. Characteristics of student study population (prior to matching).**
(DOCX)

**S2 Table. Characteristics of employee study population (prior to matching).**
(DOCX)

**S1 Fig. Absolute standardized mean difference between vaccination groups among students.** Absolute standardized mean difference between unvaccinated versus booster (left) and unvaccinated versus fully vaccinated (right) for baseline covariates before and after propensity score matching among students. Vertical lines represent the thresholds of 0.1 and 0.25.
(TIF)

**S2 Fig. Absolute standardized mean difference between vaccination groups among employees.** Absolute standardized mean difference between unvaccinated versus booster (left) and unvaccinated versus fully vaccinated (right) for baseline covariates before and after propensity score matching among employees. Vertical lines represent the thresholds of 0.1 and 0.25.
(TIF)

**S3 Fig. Histogram of propensity scores of the unmatched sample and the matched sample for students.** The common reference group (boosted) is represented in red, the unvaccinated group is represented in grey, and the fully vaccinated group is represented in green,

respectively.
(TIF)

**S4 Fig. Histogram of propensity scores of the unmatched sample and the matched sample for employees.** The common reference group (fully vaccinated) is represented in green, the unvaccinated group is represented in grey, and the boosted group is represented in red, respectively.
(TIF)

**S5 Fig. Distribution of receipt of the 2nd dose and the booster dose by calendar month for university students and employees in the matched sample.**
(TIF)

**S1 Checklist. STROBE checklist for cohort studies.**
(DOCX)

## Acknowledgments

We thank the Clemson University administration, medical staff, and all other testing providers who helped implement and manage SARS-CoV-2 testing at Clemson University. We thank Clemson's Computing & Information Technology department for their role in collecting, managing, and distributing test results.

## Author Contributions

**Conceptualization:** Lior Rennert.

**Data curation:** Lior Rennert, Delphine Dean.

**Formal analysis:** Lior Rennert, Zichen Ma, Christopher S. McMahan.

**Funding acquisition:** Lior Rennert, Delphine Dean.

**Investigation:** Lior Rennert, Zichen Ma.

**Methodology:** Lior Rennert, Zichen Ma, Christopher S. McMahan.

**Project administration:** Lior Rennert.

**Resources:** Lior Rennert.

**Software:** Lior Rennert, Zichen Ma.

**Supervision:** Lior Rennert.

**Validation:** Lior Rennert.

**Visualization:** Lior Rennert.

**Writing – original draft:** Lior Rennert.

**Writing – review & editing:** Lior Rennert, Zichen Ma, Christopher S. McMahan, Delphine Dean.

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
