## [Decision Letter · Decision Letter 0]

4 Aug 2022

PGPH-D-22-00917

Covid-19 vaccine effectiveness against general SARS-CoV-2 infection from the omicron variant: A retrospective cohort study

Dear Dr. Lior Rennert

Thank you for submitting your manuscript to PLOS Global Public Health. After careful consideration, we feel that it has merit but does not fully meet PLOS Global Public Health’s publication criteria as it currently stands. Therefore, we invite you to submit a revised version of the manuscript that addresses the points raised during the review process.

Please ensure the methodological section (selection process and statistical analysis) is clarified to allow correct data interpretation.

We look forward to receiving your revised manuscript.

Kind regards,

Prof. Patrick DMC Katoto

Academic Editor

Journal Requirements:

2. Please provide separate figure files in .tif or .eps format.

3. Since your data is not available for proprietary reasons, please explain via email why the data is not available. Please also include the contact information for the third party organization that should be contacted should other researchers want to request access to this data and please include the full citation of where the data can be found. We also request that you verify with us via email that any researcher will be able to obtain the data set in the same manner that the you have obtained it. If you feel you are unwilling or unable to adhere to this policy, please explain your reasons by return email and your exemption request will be escalated to the editor for approval. Your exemption request will be handled independently and will not hold up the peer review process, but will need to be resolved should your manuscript be accepted for publication. One of the Editorial team will be in touch if they require more information.

Additional Editor Comments (if provided):

Please ensure draft is numbered.

Reviewers' comments:

Reviewer's Responses to Questions

**Comments to the Author**

1. Does this manuscript meet PLOS Global Public Health’s publication criteria? Is the manuscript technically sound, and do the data support the conclusions? The manuscript must describe methodologically and ethically rigorous research with conclusions that are appropriately drawn based on the data presented.

Reviewer #1: Yes

Reviewer #2: Yes

2. Has the statistical analysis been performed appropriately and rigorously?

Reviewer #1: I don't know

Reviewer #2: No

3. Have the authors made all data underlying the findings in their manuscript fully available (please refer to the Data Availability Statement at the start of the manuscript PDF file)?

Reviewer #1: Yes

Reviewer #2: No

4. Is the manuscript presented in an intelligible fashion and written in standard English?

Reviewer #1: Yes

Reviewer #2: No

5. Review Comments to the Author

Reviewer #1: I did not find it easy to understand the model chosen to select or exclude participants.

I also fail to understand how in a follow-up period between 3 and 31 January 2022, it has between 27 to 30 tests performed, from which I assume that the follow-up was longer but it is not clear from the text.

I also find remarkable the low number of tobacco users, about 2%, which makes me doubt the demographic data (maybe it is a real data, if so, I think it is fantastic).

only polymerase chain reaction was used for diagnosis? at the time of the study there were many antigen self-tests available which may have also altered the possible interpetation of results. This is not mentioned on the text

Another point that seems to me of utmost importance and that is mentioned in the discussion is the reporting error that may exist in those who did receive a booster, since there is no economic incentive, so many participants could have had a booster and not be reported, which alters the interpretation of the data.

Reviewer #2: Abstract

SARS-CoV-2? Write in full then after abbreviate.

COVID-19? Write in full then after abbreviate.

“Unvaccinated” should be written “unvaccinated”

within “the” previous 5 months

Intervention should be well described, what were the BNT162b2 or mRNA-1273 vaccine doses?

Results: How many university students and employees did you include in your results? sex and age distribution should be included.

The result is poorly reported. Please include the BNT162b2 and mRNA-1273 vaccine effectiveness in the result.

Conclusion: your conclusion should be based on the study population. What are your recommendation based on your results?

Introduction

Syndrom should be written “syndrome”

Omicron has caused “a” record number of daily….

The epidemiology of COVID-19 in the US is poorly described.

Please briefly review the vaccine effectiveness of Wuhan, beta, and delta variants in the US. Compared this with recent studies conducted about Omicron variant.

Please state clearly the objective of this study and how this study can be considered as particular compared to other studies conducted in the US about Omicron variant.

we assess “the” effectiveness of the 2-dose and 3-dose messenger RNA (mRNA) vaccination

“Adminstration” should be written “administration”

Methods

The method should include a list of all sociodemographic and covariate variables used in the results.

“Time-varying Cox proportional hazard models were used to estimate the relative risk (RR)”, this use to estimate the hazard risk rate.

I would like you to compute non adjusted and adjusted RR to estimate the vaccine effectiveness.

As the p-value and the 95%CI are included in the results, this should be well stated in the statistical analysis.

Kind state in the methods that 1-aRR was used to compute the vaccine effectiveness.

Be consistent between COVID-19 and Covid-19 throughout the study.

I do not see anything about data collection and management. Please specify.

Results

As stated in the abstract, the socio-demographic characteristics should be included in your results.

Age should be classified because the Anova of unvaccinated, fully vaccinated, and booster was statistically significant. It will also be interesting to see how vaccine effectiveness varies by age distribution in this population.

As stated in the methodology, adjusted RR should be used.

Discussion

The discussion should be strengthened. How do you compare this study results to other studies related to Omicron vaccine effectiveness?

Authors just state the study limitations. Where are the study strengths?

Clearly state how misclassification of vaccination status may bias estimates of vaccine effectiveness.

Conclusion

The study recommendation for this specific population should be clearly stated.

References

Pubmed should be used for referencing. Many of the references are incorrect.

6. PLOS authors have the option to publish the peer review history of their article (what does this mean?). If published, this will include your full peer review and any attached files.

**Do you want your identity to be public for this peer review?** For information about this choice, including consent withdrawal, please see our Privacy Policy.

Reviewer #1: No

Reviewer #2: **Yes: **Jacques L Tamuzi

---

## [Decision Letter · Decision Letter 1]

8 Nov 2022

PGPH-D-22-00917R1

Covid-19 vaccine effectiveness against general SARS-CoV-2 infection from the omicron variant: A retrospective cohort study

Dear Dr. Rennert,

Thank you for submitting your manuscript to PLOS Global Public Health. Your work has received thorough consideration from our reviewers and may be published subject to the satisfaction of the minor suggestions attached. Therefore, we invite you to submit a revised version of the manuscript that addresses the points raised during the review process.

We look forward to receiving your revised manuscript.

Kind regards,

Prof Patrick DMC Katoto, MD, MSC, PhD

Academic Editor

Journal Requirements:

Reviewers' comments:

Reviewer's Responses to Questions

**Comments to the Author**

1. If the authors have adequately addressed your comments raised in a previous round of review and you feel that this manuscript is now acceptable for publication, you may indicate that here to bypass the “Comments to the Author” section, enter your conflict of interest statement in the “Confidential to Editor” section, and submit your "Accept" recommendation.

Reviewer #1: All comments have been addressed

Reviewer #2: All comments have been addressed

2. Does this manuscript meet PLOS Global Public Health’s publication criteria? Is the manuscript technically sound, and do the data support the conclusions? The manuscript must describe methodologically and ethically rigorous research with conclusions that are appropriately drawn based on the data presented.

Reviewer #1: Yes

Reviewer #2: Partly

3. Has the statistical analysis been performed appropriately and rigorously?

Reviewer #1: I don't know

Reviewer #2: Yes

4. Have the authors made all data underlying the findings in their manuscript fully available (please refer to the Data Availability Statement at the start of the manuscript PDF file)?

Reviewer #1: Yes

Reviewer #2: Yes

5. Is the manuscript presented in an intelligible fashion and written in standard English?

Reviewer #1: No

Reviewer #2: Yes

6. Review Comments to the Author

Reviewer #1: dont use Spring or fall "...in a large public university student and employee population undergoing mandatory

weekly testing during the Spring 2022 semester"

do not use spring or autumn as a measure of a time period. It is only used in the northern hemisphere and is confusing for those who do not live in that hemisphere. Use the months of the year, which are the same for everyone.

it is possible to understand, but difficult to do so, the criteria of inclusion and exclusion of the participants, I suggest to improve the wording.

How can it be explained that unvaccinated individuals had fewer infections than those vaccinated with two doses? "Compared to fully vaccinated (34.5%) and unvaccinated individuals (31.3%)".

it would be interesting to know which omicron subvariant was circulating at that date in that place.

Reviewer #2: The authors have worked hard to improve the manuscript. The majority of my previous comments have been well addressed. However, the discussion remains poorly described. How do you discuss the statistically significant P-values in table 1A? 1B? and the comparison between students and employees in this study, which was conducted in a university setting? Please make this clear in the discussion.

7. PLOS authors have the option to publish the peer review history of their article (what does this mean?). If published, this will include your full peer review and any attached files.

**Do you want your identity to be public for this peer review?** For information about this choice, including consent withdrawal, please see our Privacy Policy.

Reviewer #1: No

Reviewer #2: **Yes: **Jacques L. Tamuzi

---

## [Editor Report · Decision Letter 2]

13 Dec 2022

Covid-19 vaccine effectiveness against general SARS-CoV-2 infection from the omicron variant: A retrospective cohort study

PGPH-D-22-00917R2

Dear Professor *Lior Rennert*,

We are pleased to inform you that your manuscript 'Covid-19 vaccine effectiveness against general SARS-CoV-2 infection from the omicron variant: A retrospective cohort study' has been provisionally accepted for publication in PLOS Global Public Health.

Best regards,

Prof. Patrick DMC Katoto, MD, MSC, PhD

Academic Editor